# Effect of Dam Body Conformations on Birth Traits of Calves in Chinese Holsteins

**DOI:** 10.3390/ani13142253

**Published:** 2023-07-09

**Authors:** Jiayu Yang, Zhipeng Zhang, Xubin Lu, Zhangping Yang

**Affiliations:** College of Animal Science and Technology, Yangzhou University, Yangzhou 225009, China; mz120221551@stu.yzu.edu.cn (J.Y.); dx120180094@yzu.edu.cn (X.L.); yzp@yzu.edu.cn (Z.Y.)

**Keywords:** body conformation, birth traits, linear scores, cow, reproductive performance

## Abstract

**Simple Summary:**

The birth traits of calves are directly related to the economic benefits of the dairy farm. Predicting the birth traits of calves before they are born will benefit the breeding of cows and reduce the potential loss of pasture. Conformation traits are closely related to the reproduction of dairy cows. The objective of the present study was to examine the associations among birth traits and conformation traits in Chinese Holstein dairy cows, in order to provide a reference for understanding the birth traits of calves in advance by using the conformational traits of dam cows. In this study, we analyzed the relationship between female structure traits and calf birth traits by logistic regression and multiple comparisons. We found that there was a significant correlation between the two, which could predict the birth traits of calves to a certain extent by the body conformation traits of dams.

**Abstract:**

The objective of this study was to explore the effect of dam body conformations on birth traits including stillbirth, dystocia, gestation length and birth weight of Chinese Holstein calves and to provide a reference for improving cow reproductive performance. We collected phenotype data on 20 conformation traits of Chinese Holstein cows and analyzed the impact of dam conformation trait linear scores on stillbirth, dystocia, gestation length and calf birth weight. The feet angle, set of rear legs, fore udder attachment and rear attachment height traits of the dairy cows significantly affected the risk of stillbirth. The risk of dystocia decreases with the increase in stature and pin width. The bone quality of dairy cows had a significant positive correlation with gestation length. There was a significant positive correlation between fore udder attachment and calf weight at birth. The birth weight of calves from cows with high body conformation traits was significantly higher than that of calves with a low composite score. These results suggest that improving the body conformation traits, especially the selection of mammary system and body shape total score, will be beneficial in improving the reproductive performance of dairy cows.

## 1. Introduction

Calving problems affect the profitability of dairy herds by increasing veterinary and labor costs and reducing fertility and milk production [1]. Common calving problems include stillbirth, dystocia, shorter gestation length and lighter calf weight at birth. These problems are also important birth traits of calves [2]. This will cause direct economic damage to the ranch. We defined stillbirth as a calf with at least 260 days of gestation, but death within 24 to 48 h after parturition [3,4]. In addition, stillbirths often lead to decreased milk production and impaired reproduction, as well as dam survival is impaired [5,6]. After cows give birth to stillborn calves, the possibility of pregnancy is reduced and the risk of death throughout the lactation period increases [6]. Dystocia is defined as when a lying-down cow in labor experiences intermittent strong contractions and occasionally stands up or lies down for a period of 30 min or requires assisted labor [7]. Calving difficulties are associated with reduced survival rates of both the cow and the calf, as well as decreased lifespan, productivity and reproductive performance of the cow [8]. Calving difficulty can result in increasing rates of neonatal calf mortality, lower milk production and overall reducing the health of the cows. In theory, the gestation length of cattle begins from the formation of the fertilized egg to the time of delivery. However, due to the difficulty in determining the exact time of fertilization of the egg, technicians perform rectal examinations to estimate the expected date of delivery and calculate the gestation length based on the actual delivery time [9]. Shorter gestation length cows are more likely to calve earlier and present for mating earlier in the season, but a very short gestation length may result in stillbirth or a decline in the offspring’s condition [10]. The average gestation period of Holstein cows is 276 d. When the gestation period is 10 d or more below the average, it increases the incidence of stillbirth, retained placenta and uterine inflammation [11]. In previous studies, calf weight was associated with obstetrical assistance [12], lameness events [13] and milk production [14].

Maternal factors are the most important factors affecting the birth traits of calves. In previous studies, unfavorable heterosis was reported for direct effects in calving ease [15] and neonatal calf mortality [15,16], while favorable heterosis was reported for maternal effects in calving ease and neonatal calf mortality [15,16]. Xue told us that the heritability range of conformation traits was 0.11 (angularity) to 0.37 (heel depth) and that traits such as body height, chest width and heel depth were moderately heritable traits [17]. This suggests that dam body conformations are likely to be passed on to calves and this is likely to directly affect the birth traits of the calves. However, to the best of our knowledge, it has not been well reported to date, with only one study evaluating the impact of body conformation on dystocia [18].

Body conformation traits are important components to be considered in breeding objectives as indicators of cow efficiency. There is a genetic correlation between improved reproductive performance and body shape characteristics [19]. Cows with moderate body conditions after their first calving are more likely to survive compared to cows with high or low body conditions [20,21]. This suggests that different body types of cows may have an impact on the overall reproductive performance of the herd [20]. A body condition score assessment evaluates the energy reserve of dairy cows and is, therefore, related to their energy balance status and reproductive ability. Previous studies used multiple analyses to estimate the genetic correlation between reproductive ability and body condition quality and demonstrated that cows with a low body condition score tend to have poorer reproductive ability [22].

Body conformation traits can be evaluated by linear scoring. The linear scoring system objectively describes the conformation from one biological extreme to another, with each trait describing a specific part of the cow [23]. If the relationship between linear traits and birth traits is determined, farmers can easily select breeding cattle through visual inspection. For resource-poor farmers, the use of visual evaluation of linear traits will be more cost effective. The aim of this study is to determine whether dam body conformation affects birth traits including stillbirth, dystocia, gestation length and calf weight at birth.

## 2. Materials and Methods

### 2.1. Data Sources

The data for this study were collected from a standardized management farm in Jiangsu Province, China. The cows were sourced from the same ranch, in order to minimize environmental effects on research outcomes to the greatest extent possible. The farm adopts a double-row barn for free-range grazing of cows. They are fed with a total mixed ration (TMR) three times a day and standardized milking and calving practices are implemented. The farm also has strict feeding management procedures. The annual number of cattle in the herd is over 3800 head. Artificial insemination is employed in the ranch and the 10 types of bull semen used are sourced from a single company with similar physical characteristics and conditions. Finally, from 2020 to 2021, we screened 1474 cows of similar age, whose linear scores were measured 30 to 180 d after the first calving and who became pregnant after subsequent artificial insemination. The sample size of 1474 cows can effectively reduce the intragroup error and meet the requirements of subsequent statistical analysis [24]. The measurement of the body conformation traits was completed within three days. Furthermore, an evaluation was conducted on linear features using a nine-point scale. These linear scores were assessed with reference to a code of practice of type classification in Chinese Holstein. For each physical trait, the maximum and minimum values of the trait are determined based on the range of biological variation and then scored on a linear scale. We measured the recovery of cows to a normal physical state between 30 to 180 d of postpartum and collected the birth characteristics of the second parity. In this study, the body structure traits of all cows were determined within 30 to 180 d after lactation to exclude the influence of lactation days on the analysis. A large number of relevant studies have shown that the best time for the measurement of body structure traits is 30 to 180 d of lactation and the measured data during this period will not be affected by the number of lactation days, which is also included in the determination standard of body structure traits of Chinese Holstein cattle referred to in this study [25,26]. Table 1 shows the full names and abbreviations of the 20 linear traits of the body conformation traits. Appendix A explains the measurement criteria for converting linear scores into functional scores. The functional score is there to help account for traits where there is an intermediate optimum. When the functional score is at a higher level, the cow’s body is in a better state.

The four birth traits studied were stillbirth, dystocia, gestation length and calf weight at birth. All indicators are measured by professional veterinary technicians. After fertilization is completed, technicians will observe the fertilization process daily and whether it has been implanted. They will also calculate the expected calving date. On the expected calving date, technicians will observe the cow every one to two hours to determine if she has started calving. Stillbirth is a calf with at least 260 d of gestation, but death within 24 to 48 h after parturition. Dystocia is defined when a lying-down cow in labor experiences intermittent strong contractions and occasionally stands up or lies down for a period of 30 min or requires assisted labor [7]. Gestation length was estimated by a technician through rectal examination and adjusted for the actual time of delivery. The weight of calves before they first eat colostrum is their birth weight. The birth weight of calves is usually measured with a scale. After the calf is born, after the mucus in the mouth and nose of the calf has been treated and the umbilical cord has been severed, two people work together with one lifting the front legs and one lifting the hind legs, and the calf is placed on a scale for weight measurement. Values for the body conformation and birth traits were determined by a professional assessor. The measurement standards of conformation traits were based on the “Code of practice of type classification in Chinese Holstein” (https://std.samr.gov.cn/gb/gbQuery, Standard No: GBT35568-2017 accessed on 1 July 2018). Stillbirths and dystocia were coded as categorical traits (1 = had stillbirth, 0 = did not have stillbirth; 1 = had dystocia, 0 = did not have dystocia).

### 2.2. Statistical Analyses

We used SPSS statistical software (SPSS 26.0, IBM, Ehningen, Germany) to analyze and process the data. Using logistic regression to analyze the effect of ST, CW, BD, LS, PS, PW, FA, HD, BQ, SRL, RLRV, UD, MS, FUA, FTP, FUL, RAH, RAW, RTP and ANG on stillbirths or dystocia in cattle, the model is as follows:$$\begin{array}{c} Logit\;\left(p\right)=\ln(\frac{p}{1-p})\\ =\beta_{\gamma }+\beta_{1}{ST}_{i}+\beta_{2}{CW}_{j}+\beta_{3}{\;BD}_{k}+\beta_{4}{LS}_{l}+\beta_{5}{PS}_{m}+\beta_{6}{PW}_{n}+\beta_{7}{FA}_{o}+\beta_{8}{HD}_{p}+\beta_{9}{BQ}_{q}\\ +\beta_{10}{SRL}_{r}+\beta_{11}{RLRV}_{s}+\beta_{12}{UD}_{t}+\beta_{13}{MS}_{u}+\beta_{14}{FUA}_{v}+\beta_{15}{FTR}_{w}+\beta_{16}{FUL}_{x}+\beta_{17}{RAH}_{y}\\ +\beta_{18}{RAW}_{z}+\beta_{19}{RTP}_{a}+\beta_{20}{ANG}_{b} \end{array}$$
where *p* represents whether the calf was a stillbirth or the cow had dystocia, *β_γ_* is the constant term, *STi* is the fixed effect of stature, *CW_j_* is the fixed effect of chest width, *BD_k_* is the fixed effect of body depth, *LS_l_* is the fixed effect of loin strength, *PS_m_* is the fixed effect of pin setting, *PW_n_* is the fixed effect of pin width, *FA_o_* is the fixed effect of feet angle, *HD_p_* is the fixed effect of heel depth, *BQ_q_* is the fixed effect of bone quality, *SRL_r_* is the fixed effect of the set of rear legs, *RLRV_s_* is the fixed effect of rear leg-rear view, *UD_t_* is the fixed effect of udder depth, *MS_u_* is the fixed effect of median suspensory, *FUA_v_* is the fixed effect of fore udder attachment, *FRP_w_* is the fixed effect of fore teat placement, *FUL_x_* is the fixed effect of fore udder length, *RAH_y_* is the fixed effect of rear attachment height, *RAW_z_* is the fixed effect of rear attachment width, *RTP_a_* is the fixed effect of rear teat placement and *ANG_b_* is the fixed effect of angularity. The *β1–β20* are regression coefficients for each effect.

The odds ratio (OR) reflects the strength of the association between each unit of linear rating and the occurrence of stillbirth or dystocia. OR was calculated as the ratio of the probability of stillbirth or dystocia for one linear score for a trait to the probability of stillbirth or dystocia for other different linear scores for that trait. The confidence interval represents the degree to which the true value of a parameter is likely to fall around the measured results. The confidence interval provides a measure of the reliability of the measured value of the parameter being tested. The level represents different linear scores.

Pearson’s correlations were used to calculate the correlation between pregnancy days, birth weight and different traits. The composite score (CS) represents the comprehensive evaluation of the cows’ body shape. The calculation of the CS utilizes the functional score, which is represented by the following formula:$$\begin{array}{c} The\;composite\;score\\ \;=18\%*\left(ST*25\%+CW*35+BD*25+LS*15\right)+10\%\\ \;*\left(PS*40\%+PW*45\%+LS*15\right)+20\%\\ \;*\left(FA*25\%+HD*15\%+BQ*15\%+SRL*25\%+RLRV*20\%\right)+42\%\\ \;*[20\%*\left(UD*55\%+MS*45\%\right)+35\%\\ \;*\left(FUA*45\%+FTP*25\%+FUL*18\%+UD*12\%\right)+45\%\\ \;*\left(RAH*30\%+RAW*30\%+RTP14\%+UD*12\%+MS*14\%\right)]+\\ \;10\%*(ANG*80\%+BQ*20\%) \end{array}$$

We classified the comprehensive scores as follows: 75–79 points as good (G), 80–84 points as very good (VG) and 85–89 points as excellent (EXC). The classification and calculation of the CS are according to the “Code of practice of type classification in Chinese Holstein“. However, as only data with a CS of 85–86 were collected, we defined this range as excellent (EXC). We used the mean plus or minus the standard deviation to represent the pregnancy days of cows and the birth weight of calves under different traits and scores.

Duncan’s multiple range test was used for multiple comparisons to test the impact and specific differences of cow pregnancy days and calf birth weight under different linear scores for different conformation traits.

## 3. Results

### 3.1. Descriptive Statistics of Body Conformation Traits 

In Table 2, we present the mean score of the linear scores for 20 body conformation traits, which ranged from 3.04 (SRL) to 7.93 (ST). The mean score for CS was 80.5. The standard deviation ranged from 0.49 (FTP) to 1.18 (SRL) and the standard deviation for CS was 1.96. The distribution of the body conformation traits in the herd is shown in Table 3.

### 3.2. Effect of Body Conformation Traits of Cow on Stillbirth

Table 4 shows as the linear scores of FA and SRL increase in cows, the probability of stillbirth in cows also increases. When the FA of the cow was 5 points, the risk of stillbirth was the lowest relative to other linear scores of FA (OR = 0.04) and the effect of FA of 5 points compared with 4 points on the risk of stillbirth was significantly different (*p* < 0.05). Cows had the lowest risk of stillbirth relative to other points of SRL when the 2 points of SRL (OR = 0.12). The risk of stillbirth at 2 points of SRL was significantly different from that at 1 point (*p* < 0.05). When cows had 4 points for FUA, the OR value was greater than 1 point, indicating that the linear score for FUA increased the probability of stillbirth. In total, 7 points of RAH were associated with the lowest risk of stillbirth compared with other RAH linear scores (OR = 0.02). The 6 points and 7 points of RAH were associated with a significant risk of stillbirth compared with 4 points of RAH (*p* < 0.05). Linear scores for other traits for which no significance was found for stillbirth are presented in Appendix A. 

### 3.3. Effect of Body Conformation Traits of Cow on Dystocia

According to Table 5, the linear score of PW has a significant effect on the occurrence of dystocia in cows (*p* < 0.05). The risk of dystocia was lowest for a PW of 6 points (OR = 0.02) compared with a PW of 5 points. When ST was 7 points, the risk of dystocia was the lowest compared with 4 points (OR = 0.01). In Appendix A, linear scores for traits that were not significant for dystocia are presented here.

### 3.4. Association of Body Conformation Traits of Cow with Gestation Length

As shown in Table 6, there is a positive correlation between BQ and gestation length (*p* < 0.05). There was no significant difference in gestation length among the G group, VG group and EXC group cows (Figure 1A). The relationship between 20 body traits and gestation length in dairy cows is shown in Appendix A. There is no significant impact between them.

### 3.5. The Association between Body Conformation Traits and Birth Weight of Cows at Different Grades

The FUL of cows has a small positive correlation with the birth weight of calves (Table 6). CS significantly affects the birth weight of calves. Calves born to EXC group cows have significantly higher birth weights than those born to VG group cows (Figure 1B). Table 7 shows that the linear scores of SRL and RAH in cows have a significant impact on the birth weight of calves. Calves born to cows with an SRL score of 5 have a significantly higher birth weight than those born to cows with scores of 1 and 8, while calves born to cows with an RHL score of 5 have a significantly lower birth weight than those born to cows with other linear scores. 

## 4. Discussion

For dairy farms, stillbirth is an important economic feature. In this study, we observed as the linear scores of FA and SRL increased in cows, the probability of stillbirth in cows also increased. Previous research has shown that these mammary systems and feet and legs had a strong relationship with functional survival [27]. In a previous study on the effect of foot and leg conformation traits on genomic predictions of claw disorders, the results indicated that including factor analysis (FA) marginally improved the prediction correlation of breeding values for the helical claw genomic traits, both in 10-fold cross-validation (from 0.35 to 0.37) and in validation including the youngest breed (from 0.38 to 0.49) [28]. In addition, a significant correlation (0.37) between lameness and SRL has been demonstrated. Research has indicated that lameness in dairy cows can have an impact on the maintenance of pregnancy, which parallels our experimental findings [29]. Our study found that cows with an FA score of 5 had the lowest risk of stillbirth. As the angle increases, so does the risk of stillbirth. The risk of stillbirth is lowest when the SRL score of cows is 2 and it increases as the degree of hind limb flexion increases. Perhaps FA and SRL may have an impact on hoof function, which could lead to the occurrence of stillbirths. It has been proven that the shape of the udder and teats of cows and water buffaloes is significantly related to the occurrence of mastitis [18]. According to the study by Sinha et al., term udder and term udder morphology have a significant impact on the probability of subclinical mastitis in Holstein cows [30]. Animals with weaker attachment to the front mammary gland and longer posterior mammary glands appear to be more susceptible to mammary gland inflammation [31]. Meanwhile, research has found that subclinical mastitis can affect calf stillbirth rates [32]. This may be one of the reasons for the following results in our experiment. In our study, we found that the stronger the connection between the udder and the abdominal wall of a cow, the lower the occurrence of stillbirths. The ideal calving score for dairy cow RAH is 9; however, the risk of stillbirth is lowest when the score is between 6 and 7. This may be because when cows receive subclinical mastitis it affects the stillbirth rate of calves.

Dystocia, also known as difficult calving, has a deleterious effect on the health of cows and represents a negative financial burden on the dairy sector [33]. Several risk factors have been identified as associated with dystocia in cattle, including birth weight and gender of the calf [34]. However, variables such as calf sex, calf weight or the body condition of the cow during or after calving can only be determined after insemination or calving [35]. However, in our study, it was hoped that the birth traits of calves could be predicted to some extent by the body size traits of heifers. As scholars have discovered, although the weight of Belgian Blue calves does not show a correlation of 0.65 with calf leg morphology and height as reported by Freking, the correlation is still significant, which may indirectly affect the occurrence of dystocia in calves [36]. At the same time, scholars have found that ST belongs to a moderately heritable trait, so we speculate that the body condition of the cows may also have an impact on dystocia [17]. In this study, we found that ST and PW had significant effects on dystocia. Our results are in line with the study by Bila et al., which suggests that these body conformation traits may be used to predict the occurrence of dystocia in cows [37]. Therefore, in the breeding process, ST and PW can be well incorporated into selection criteria to reduce the likelihood of dystocia in heifers. In our study, due to sample size limitations, we chose to use logistic regression analysis to examine stillbirth and dystocia. In addition, we think it is more valuable to use Legendre polynomials in future research. Using Legendre polynomials as a modeling method will allow for a more comprehensive assessment of muscle and body condition, taking into account the nonlinear relationship between these characteristics and reproductive outcomes. So using Legendre polynomials will make the study more complete [38].

Effective management of livestock pregnancy and health is a crucial aspect of dairy farm operations and improving animal welfare [39]. Few studies have documented the effects of body conformation traits on gestation length. This study demonstrates that BQ may exhibit a significant positive correlation with gestation length in cows. However, this correlation is weak and may be attributed to other factors such as disease and stress which also influence the duration of pregnancy. Further research is needed to confirm these findings. The BQ reflects the thickness of the leg of the cattle. A previous study showed that BQ was associated with aggression in Charolais beef cattle during pregnancy [40]. BQ also represents durability and flexibility as well as the refinement and solidity of the hind limb bones. Perhaps it is related to exercise, which may be more conducive to natural childbirth [41]. Ferreira’s research on humans indicates that exercise during pregnancy is more beneficial for childbirth [42]. We speculate that this may also have an impact on cattle. Hansen found that when the gestation length of cows is between 275 and 280 days, there is a greater probability of cows giving birth without assistance and a lower incidence of stillbirths [43]. Other studies have also found that shortening the perinatal period during pregnancy could extend negative effects on offspring [44]. Therefore, further research is needed to determine the optimal specific body condition score to ensure that pregnancy length is not too long or too short in order to deliver at the optimal time and to reduce the incidence of stillbirth. The increase in birth weight of calves is related to dystocia, stillbirth and calf mortality, all of which are associated with reduced reproductive performance in both calves and cows and may result in economic losses [14]. In this study, we also found a small positive correlation between the FUL and the birth weight of calves. Previous research has shown that ST, CW and PW appeared to be positively correlated with calf size [45]. The CS significantly influences calf weight at birth. In this study, the birth weight of cows in EXG is significantly higher than those in the G and VG. Previous studies have shown that there is a high genetic correlation between body conformation and health traits [46]. Previous research has shown that the body condition score of cows is related to the health status of their newborn calves and that the birth weight of calves is positively correlated with their health status [47]. Regarding the results obtained in our study, we believe that the body condition score of the cows may have an impact on the birth weight of calves. In linear scoring, we consider a score of 5 for RLRV to be the most ideal score, which means it is healthier compared to other scores [48]. This is why, in our research results, calves born from cows with a score of 5 have a higher birth weight than those from cows with a score of 1 or 8. For RAH, the optimal score is 9, so in comparison, a score of 5 is considered poor. This may explain why calves born from cows with a RAH score of 5 have a lighter birth weight compared to other scores. So we think an important reason for the interesting association between dam body conformation and calf birth traits observed in this study, may be attributed to health factors. On the other hand, when the pregnancy period is longer, cows have more milk, fat and protein, which benefits farm economic benefits [49]. Additionally, the birth weight of calves is related to the pregnancy period and has a linear relationship, the longer the pregnancy period, the heavier the birth weight of the calf [50]. However, a heavier birth weight is not necessarily better because for every 1 kg increase in birth weight, the probability of difficult labor increases by 13% [12]. Therefore, further research is needed to explore if the body condition score of a cow is optimal when calf birth weight is least likely to cause difficult calving.

Although studies have shown that crossbreeding between Angus beef cattle and Holstein cows leads to faster calf growth and increased pasture income [51], purebred Holstein cattle are less prone to dystocia and have a shorter gestation length [52]. In our experiment, the semen on the ranch was selected from purebred Holstein cattle to ensure experimental consistency. Studies have shown that the selection of different bulls can affect the gestation length of cows [53] and the ease of calving [54], as well as the birth weight of calves. Although the semen we selected came from the same company to reduce the impact on pastures that do not use semen and choose to breed bulls, attention should be paid to the stress of the bulls to the new environment [38] and the damage to the cows during mating.

In practical terms, our findings will be useful for how they can be applied to dairy farming practices. Improving the selection of body size traits, especially the mammary gland system and total body size, will improve the reproductive performance of cows and can help farmers improve calf birth weight and reduce the risk of stillbirth and dystocia. These findings can help farmers make informed decisions about breeding and management strategies.

## 5. Conclusions

In conclusion, as the linear scores of FA and SRL increase in cows, the probability of stillbirth in cows also increases. Compared with other linear scores of FA, SRL and RAH, calves born to cows with a score of 5 in FA, 2 in SRL or 7 in RAH have a lower risk of stillbirth, whereas calves born to cows with a score of 8 in FUA have a higher risk of stillbirth. There was a significant difference between the linear score of PW and the risk of dystocia. BQ may have a significant positive correlation with gestation length in cows. According to Appendix A, none of the body size traits affected gestation length. We also found some positive associations between FUL and calf birth weight. The CS of the cow had an effect on the birth weight of the calf. Heifers with CS scores of 85 to 86 were significantly higher than those with CS scores of 80 to 84. SRL and RAH had an effect on calf birth weight. In cows, the birth weight of calves born to cows with an SRL score of 5 is higher than that of cows with a score of 1 and 8. Additionally, the birth weight of calves born to cows with a RAH score of 5 is lower than that of cows with scores of 6, 7 and 8. These results suggest that reducing stillbirths and dystocia can be achieved through the body structure traits of female animals. And what we found may have valuable implications for future research on the gestation length and calf birth weight and can benefit the breeding of cattle while reducing potential economic losses in the livestock industry.

## Figures and Tables

**Figure 1 animals-13-02253-f001:**
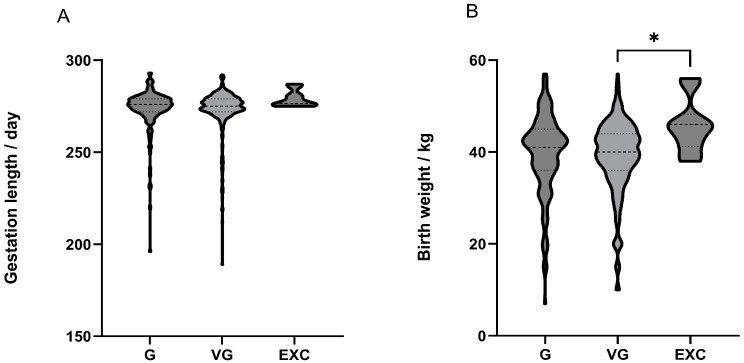
The distribution of calf birth weights and gestation lengths at different classifications of composite scores for body conformation. The classification of the comprehensive scores is 75–79 points as good (G), 80–84 points as very good (VG) and 85–86 points as excellent (EXC). Violin plots are three levels of linear total scores of adult cows for calf gestation length (**A**) and birth weight (**B**), the whiskers show data spread. * indicates *p* < 0.05.

**Table 1 animals-13-02253-t001:** Full names and abbreviations of 20 linear traits of body conformation traits.

Abbreviation	Full Name	Abbreviation	Full Name
ANG	angularity	BD	body depth
BQ	bone quality	CW	chest width
FA	feet angle	FTP	fore teat placement
FUA	fore udder attachment	FUL	fore udder length
HD	heel depth	LS	loin strength
MS	median suspensory	PS	pin setting
PW	pin width	RAH	rear attachment height
RAW	rear attachment width	RLRV	rear leg-rear view
RTP	rear teat placement	SRL	set of rear legs
ST	stature	UD	udder depth

**Table 2 animals-13-02253-t002:** Descriptive statistics of the body conformation traits of Holstein cows in China ^1^.

Trait	Mean Score	Standard Deviation	Lowest Score	Highest Score
ST	7.93	0.92	4	9
CW	4.77	0.86	3	8
BD	6.66	0.56	4	8
LS	7.04	0.72	4	8
PS	4.82	1.04	2	8
PW	6.73	0.63	4	8
FA	5.92	0.93	2	8
HD	5.81	0.96	3	8
BQ	6.38	0.82	2	8
SRL	3.04	1.18	1	8
RLRV	6.91	1.12	2	9
UD	5.06	1.04	2	8
MS	5.08	1.03	3	8
FUA	5.78	0.80	2	8
FTP	5.09	0.49	3	7
FUL	4.63	0.77	3	7
RAH	6.29	0.81	4	8
RAW	6.22	0.99	3	8
RTR	5.78	0.79	4	8
ANG	5.88	0.67	4	8
CS	80.52	1.96	75	86

^1^ ST, stature; CW, chest width; BD, body depth; LS, loin strength; PS, pin setting; PW, pin width; FA, feet angle; HD, heel depth; BQ, bone quality; SRL, set of rear legs; RLRV, rear leg-rear view; UD, udder depth; MS, median suspensory; FUA, fore udder attachment; FTP, fore teat placement; FUL, fore udder length; RAH, rear attachment height; RAW, rear attachment width; RTR, rear teat placement; ANG, angularity; CS, composite score. ST, CW, BD, LS, PS, PW, FA, HD, BQ, SRL, RLRV, UD, MS, FUA, FTP, FUL, RAH, RAW, RTR and ANG are used by linear score. CS is calculated by converting linear scores into functional scores.

**Table 3 animals-13-02253-t003:** Number of Chinese Holstein cows per linear score for each trait ^1^.

Trait	Linear Score								
	1	2	3	4	5	6	7	8	9
ST	0	0	0	6	8	82	326	614	438
CW	0	0	27	617	551	233	41	5	0
BD	0	0	0	5	25	460	965	19	0
LS	0	0	0	8	30	217	860	359	0
PS	0	11	118	444	526	307	63	5	0
PW	0	0	0	5	36	408	926	99	0
FA	0	11	25	63	244	781	334	16	0
HD	0	0	19	80	430	641	241	63	0
BQ	0	5	11	14	101	655	630	58	0
SRL	41	504	510	280	90	16	27	6	0
RLRV	0	11	14	36	52	320	532	504	5
UD	0	8	69	307	690	277	87	36	0
MS	0	0	47	455	416	469	68	19	0
FUA	0	5	5	55	400	803	189	17	0
FTP	0	0	8	82	1167	206	11	0	0
FUL	0	0	16	726	540	164	28	0	0
RAH	0	0	0	55	140	627	630	22	0
RAW	0	0	27	58	195	532	602	60	0
RTR	0	0	0	11	617	553	279	14	0
ANG	0	0	0	14	375	860	219	6	0

^1^ ST, stature; CW, chest width; BD, body depth; LS, loin strength; PS, pin setting; PW, pin width; FA, feet angle; HD, heel depth; BQ, bone quality; SRL, set of rear legs; RLRV, rear leg-rear view; UD, udder depth; MS, median suspensory; FUA, fore udder attachment; FTP, fore teat placement; FUL, fore udder length; RAH, rear attachment height; RAW, rear attachment width; RTR, rear teat placement; ANG, angularity.

**Table 4 animals-13-02253-t004:** The body conformation traits of cow that have a significant impact on stillbirths ^1^.

Trait	Level	*p* Value	OR Value	95% Confidence Interval	
				Lower Limit	Upper Limit
FA	5 points vs. 4 points	0.03	0.04	0.00	0.71
	6 points vs. 4 points	0.11	0.10	0.01	1.67
	7 points vs. 4 points	0.19	0.11	0.00	3.08
SRL	2 points vs. 1 point	0.03	0.12	0.02	0.83
	3 points vs. 1 point	0.11	0.18	0.02	1.50
	4 points vs. 1 point	0.19	0.21	0.02	2.20
	5 points vs. 1 point	0.35	0.15	0.00	8.54
	6 points vs. 1 point	0.45	5.12	0.07	361.02
FUA	5 points vs. 4 points	0.73	1.59	0.11	22.88
	6 points vs. 4 points	0.56	2.31	0.14	37.81
	7 points vs. 4 points	0.31	5.24	0.21	128.91
	8 points vs. 4 points	0.05	626.55	1.09	359,182.88
RAH	5 points vs. 4 points	0.08	0.05	0.00	1.46
	6 points vs. 4 points	0.03	0.04	0.00	0.76
	7 points vs. 4 points	0.02	0.02	0.00	0.59
	8 points vs. 4 points	0.71	2.47	0.02	311.39

^1^ FA, feet angle; SRL, set of rear legs; FUA, fore udder attachment; RAH, rear attachment height; OR, the odds ratio. The FA scores range from 4 to 7 points. The SRL scores range from 1 to 6 points. The FUA scores range from 4 to 8 points. The RAH scores range from 4 to 8 points. The lowest linear score collected to produce stillbirth for each conformational trait was used as the reference level for that conformational trait construct.

**Table 5 animals-13-02253-t005:** Body conformation traits of cow that have a significant impact on dystocia ^1^.

Trait	Level	*p* Value	OR Value	95% Confidence Interval	
				Lower Limit	Upper Limit
ST	5 points vs. 4 points	0.10	0.03	0.00	2.04
	6 points vs. 4 points	0.07	0.02	0.00	1.41
	7 points vs. 4 points	0.03	0.01	0.00	0.69
	8 points vs. 4 points	0.08	0.02	0.00	1.50
PW	6 points vs. 5 points	0.01	0.02	0.00	0.43
	7 points vs. 5 points	0.02	0.03	0.00	0.52
	8 points vs. 5 points	0.02	0.03	0.00	0.61

^1^ ST, stature; PW, pin width; OR, the odds ratio. The ST scores range from 4 to 8 points. The PW scores range from 5 to 8 points. The lowest linear score collected to produce stillbirth for each conformational trait was used as the reference level for that conformational trait construct.

**Table 6 animals-13-02253-t006:** Correlations of body conformation traits of cow with gestation length and birth weight ^1^.

Trait	Gestation Length	Birth Weight
ST	0.009	0.019
CW	−0.023	−0.012
BD	0.015	−0.018
LS	−0.010	0.006
PS	−0.053	−0.011
PW	0.042	0.000
FA	−0.055	−0.073
HD	−0.010	0.017
BQ	0.091 *	0.038
SRL	−0.008	0.032
RLRV	0.038	0.006
UD	0.017	0.050
MS	−0.022	−0.029
FUA	−0.032	−0.030
FTP	0.052	0.003
FUL	0.069	0.088 *
RAH	0.045	0.011
RAW	−0.018	−0.062
RTP	−0.046	−0.056
ANG	0.037	0.037

^1^ ST, stature; CW, chest width; BD, body depth; LS, loin strength; PS, pin setting; PW, pin width; FA, feet angle; HD, heel depth; BQ, bone quality; SRL, set of rear legs; RLRV, rear leg-rear view; UD, udder depth; MS, median suspensory; FUA, fore udder attachment; FTP, fore teat placement; FUL, fore udder length; RAH, rear attachment height; RAW, rear attachment width; RTR, rear teat placement; ANG, angularity. * correlation is significant at the 0.05 level. Pearson’s correlations are used in Table 6.

**Table 7 animals-13-02253-t007:** The mean birth weight (kg) by linear score for each body conformation trait ^1^.

Trait	1	2	3	4	5	6	7	8	9	*p* Value
ST				40.00 ± 2.00	42.33 ± 3.93	38.00 ± 1.57	39.82 ± 0.81	38.68 ± 0.54	39.88 ± 0.61	0.59
CW			38.70 ± 3.59	39.40 ± 0.57	39.00 ± 0.55	40.11 ± 0.78	38.53 ± 2.34	26.50 ± 16.50		0.29
BD				31.00 ± 21.00	41.89 ± 1.84	39.44 ± 0.67	39.23 ± 0.42	36.71 ± 3.90		0.44
LS				38.67 ± 1.33	38.64 ± 1.78	39.92 ± 0.77	38.98 ± 0.49	39.66 ± 0.70		0.86
PS		35.50 ± 6.64	40.42 ± 1.12	38.86 ± 0.67	39.48 ± 0.56	39.58 ± 0.79	38.26 ± 1.66	30.50 ± 10.50		0.54
PW				33.50 ± 13.50	38.38 ± 1.39	39.83 ± 0.67	39.03 ± 0.46	39.03 ± 1.07		0.66
FA		45.00 ± 2.12	41.22 ± 1.82	39.35 ± 1.65	39.48 ± 0.85	39.39 ± 0.45	38.86 ± 0.86	32.33 ± 4.90		0.30
HD			35.43 ± 4.70	38.17 ± 1.66	40.13 ± 0.58	38.83 ± 0.55	39.22 ± 0.93	40.74 ± 1.46		0.39
BQ		36.50 ± 10.50	39.00 ± 4.74	35.20 ± 3.41	38.68 ± 1.69	39.27 ± 0.53	39.50 ± 0.51	39.19 ± 2.04		0.93
SRL	32.53 ± 2.43 ^b^	39.48 ± 0.59 ^ab^	39.35 ± 0.59 ^ab^	39.44 ± 0.84 ^ab^	41.24 ± 1.19 ^a^	38.00 ± 4.71 ^ab^	38.20 ± 2.92 ^ab^	31.50 ± 5.50 ^b^		0.04
RLRV		40.50 ± 4.13	42.60 ± 1.57	41.54 ± 2.15	35.89 ± 2.21	39.17 ± 0.71	39.08 ± 0.61	39.60 ± 0.60	41.00 ± 2.00	0.58
UD		41.33 ± 1.45	38.8 ± 1.99	38.67 ± 0.69	39.03 ± 0.55	40.53 ± 0.78	39.22 ± 1.20	40.08 ± 2.05		0.73
MS			41.65 ± 2.09	39.15 ± 0.64	40.09 ± 0.61	38.18 ± 0.67	40.00 ± 1.30	43.14 ± 3.65		0.17
FUA		39.50 ± 0.50	34.00 ± 5.00	39.90 ± 1.60	39.84 ± 0.64	39.11 ± 0.50	38.64 ± 0.97	40.67 ± 4.51		0.88
FTP			33.33 ± 6.69	41.77 ± 1.27	39.07 ± 0.41	39.45 ± 0.82	43.25 ± 1.70			0.23
FUL			42.33 ± 1.26	38.40 ± 0.53	40.01 ± 0.55	39.90 ± 1.02	42.50 ± 1.46			0.12
RAH				36.45 ± 2.16 ^ab^	41.04 ± 0.83 ^b^	38.86 ± 0.54 ^a^	39.74 ± 0.54 ^a^	33.63 ± 4.53 ^a^		0.04
RAW			40.80 ± 2.09	40.05 ± 2.15	40.56 ± 0.86	39.23 ± 0.58	38.69 ± 0.57	40.00 ± 1.70		0.61
RTR				41.00 ± 1.87	39.78 ± 0.53	39.15 ± 0.58	38.05 ± 0.85	45.20 ± 3.04		0.20
ANG				38.60 ± 4.19	39.60 ± 0.75	38.68 ± 0.47	41.11 ± 0.71	39.00 ± 2.00		0.20

^1^ ST, stature; CW, chest width; BD, body depth; LS, loin strength; PS, pin setting; PW, pin width; FA, feet angle; HD, heel depth; BQ, bone quality; SRL, set of rear legs; RLRV, rear leg-rear view; UD, udder depth; MS, median suspensory; FUA, fore udder attachment; FTP, fore teat placement; FUL, fore udder length; RAH, rear attachment height; RAW, rear attachment width; RTR, rear teat placement; ANG, angularity. Plus or minus standard deviation was used for analysis. a, b indicates a significant difference within the group. ab indicates no significant difference within the group. Table 7 employs Duncan’s multiple range test. There are no cows with the linear score for the trait when cells are blank.

## Data Availability

The data presented in this study are available in article or Appendix A.

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
