# Peer review of "Effect of Dam Body Conformations on Birth Traits of Calves in Chinese Holsteins"

_animals, 2023, doi:10.3390/ani13142253_

Round 1

Reviewer 1 Report

Line 14: Where did you do predictions of birth weight? From what I can tell, you just looked at correlations between calf and cow traits and regression analyses.

Line 22: Delete comma after cows.

Lines 28-29: Did you measure the body conformation on the calves or the cows? If on the calves, fine as is. If on the cows, this needs to say "calves from cows" not just "calves".

Line 39: "stillbirth" instead of "Stillbirth"; "as" instead of "is"

Line 40: "days" is usually abbreviated as "d"

Line 42: What about dam survival? Is it impaired? Improved?

Line 43: How long is too long?

Line 45: "mother cow" is redundant. To be a "cow", the cow must be a mother. Otherwise, it is a heifer, steer, or bull.

Lines 51-53: shorter, average, longer gestation lengths: What are the ranges? What does the number in parentheses mean? Is this an average or a cutoff? If it's a cutoff, how do you have a cutoff for average?

Lines 56-61: Combine sentences into one. "In previous studies, unfavorable heterosis was reported  for direct effects in calving ease [14] and neonatal calf mortality [14,15] while favorable heterosis was reported for maternal effects for calving ease and neonatal calf mortality [14,15]."

Line 61: What do you mean by genetic range? Are you referring to heritabilities?

Lines 62-63: Unnecessary capitalized letters (Angularity, Heel, Body, Chest).

Line 63: What do you mean by medium genetic traits? Are you talking about moderately heritable traits?

Line 69: "adjusting for milk yield"; What traits? The body conformation traits?

Lines 70-71: Improper use of comma - have an unnecessary comma and missing commas around "therefore"

Line 72: "reproductive ability"

Lines 77-79: Make this clearer. Here you say that Table S1 shows the relationship between linear scores (i.e. 9-point scale) and performance but on Table S1, you say that it is the relationship between linear and functional scores. I assume that the functional score is there to help account for traits where there is an intermediate optimum.

Line 82-82: "dam body conformation affects birth traits" - each dam only has one conformation

Line 87: "for" instead of "of"

Line 89: "were" instead of "are"; What do you mean by in the same lactation cycle?

Line 94: pretty sure there is a double space before "fore"

Lines 90-95: May be better to present in tabular form and alphabetical by abbreviation to make it easier for the reader to refer back to the abbreviations.

Line 95: Within three days of what?

Line 96: How do the structural characteristics differ from the 20 body conformation traits?

Line 98: Why such a range in days post-parturition for determining conformation? Wouldn't that impact some of your measures?

Line 99: You defined stillbirth in the introduction but how did you determine if a cow had dystocia? Was there a time limit for her to calve? Did you have 24-h monitoring to ensure you knew when calving began?

Line 99: How did you determine gestation length? I.e. how did you determine when the fertilized egg was formed?

Lines 99-100: When were calves weighed? Did you have 24-h monitoring to ensure that calves were weighed within a certain time frame?

Line 106: "had/have dystocia" - no "an"

Line 110: "technical"

Line 111: What is the comprehensive score? This has not been described previously.

Line 112-113: Abbreviations don't make sense based on description. Why does VG stand for excellent instead of very good? G, VG, and EXC would make more sense. How did you determine the point ranges for each of the categories? Did you have any cows that fell below G?

Line 116: Should be "Duncan's multiple range test"

Line 118: Should be "Kendall rank correlations were". Also, why would you use a rank correlation here? If several cows have the same ST score, it would be random what rank they are given and that order can significantly impact your results. A Pearson correlation would be much more appropriate. Also, are you using the linear or functional score for the body composition traits?

Lines 119-121: Looks like a footnote to a table was copied here by accident.

Lines 125 & 135: "calf was stillborn or the cow had dystocia"

Line 126: Should be "regression coefficient for composite score"

Lines 127-131 and 136-145: Use the abbreviations that you defined in lines 90-95.

Line 131: "on stillbirths or dystocia"

Methods: You have OR value in your tables but don't have in your methodology what that is or how it was calculated.

Table 1: All abbreviations must be defined at first usage in the abstract, main text, and in figures/tables. What is this table actually showing us? What is the OR value? What do the different things under "Level" mean? What is the 95% confidence interval of?

Lines 150-155: I'm not sure how Table 1 shows this.

Lines 155-157: Table S2 does not show the results of the regression analyses. If it were showing the results of the regression analyses, it would be giving the betas that you would have had to have estimated in your work (i.e. equations on lines 124 and 132-134). Additionally, in Table S2, under "Level", there is no space between "vs" and the second number.

Section 3.2: very similar comments to section 3.1 except pertaining to Table 2 and S3.

Table 3: The table is not showing the "association"; it is showing the "Correlations"

Section 3.3 and 3.4: While there are significant correlations, these correlations are still extremely weak correlations and do not show a relationship of body conformation traits with gestation length or calf birth weight. However, this must be taken lightly considering my concerns with using Kendall rank versus Pearson correlations.

Line 185: CS? I assume this is comprehensive score but it has not been defined prior to being used.

Lines 185-187: Why bother defining these ranges as good, very good, and excellent if you are just going to refer to the point ranges here instead of using the classifications you defined in your methodology?

Picture 1: Usually needs to be "Figure" not "Picture". This figure is also not showing the "relationship". A more appropriate caption would be "The distribution of calf birth weights and gestation lengths at different classifications of comprehensive scores for body conformation." Remember that tables and figures should be able to stand alone. That also means no usage of abbreviations without defining them. In footnote, you have that the * means the correlation is significant at the 0.05 level. However, this figure doesn't show correlations.

Lines 191-192: Table S4 does not show the relationship between body conformation and gestation length. It shows the mean gestation length (d) by linear score for each body conformation trait. There should also be a note on the table indicating that you did not have cows with the linear score for the trait when cells are blank. Because of Tables S4 and 4, it would be nice to have a supplemental table that shows the number of cows in each of the linear score categories for each body conformation trait. Also, I assume the character after "P" means value but it needs to be changed from the Chinese character.

Header: Somehow starting on page 7, the header is messed up. There is no page number on page 7 or 8 and then page 9 says it is 2 of 12.

Table 4: It does not show the relationship between birth weight and body conformation traits. It shows the mean calf birth weight (kg) by linear score for each body conformation trait. Relationship usually indicates correlations and these values are most definitely not correlations although there is no unit anywhere in the table to identify these values. Also, I assume the character after "P" means value but it needs to be changed from the Chinese character.

Lines 215-216: You may have data that supports this statement but what you have presented in the results does not.

Lines 216-225: What does this have to do with your study? Has someone tied lameness and claw health to the birth traits you evaluated?

Lines 216-218: Need citations.

Line 218: "foot"

Line 219: "the"

Line 224: two spaces between "welfare" and "and"

Line 226: "angle"

Line 228-229: This is pretty far reaching conclusion based on no evidence. Your evidence is weak at best from what is shown in this manuscript that FA and SRL are tied to incidence of stillbirths and you have shown no previous research that relates FA and SRL to incidence of hoof disease.

Lines 229-238: How does mastitis relate to your study? Has there been research that shows that mastitis is related to stillbirth, dystocia, gestation length, or calf birth weight? What are you trying to get at with this discussion on mastitis?

Line 231: Remove "(2022)"

Line 242: Delete comma after birth weight

Lines 246 & 250: "As some scholars" but only one citation? Same question for lines 250 & 252.

Lines 250-251: "moderate genetic trait"??? Do you mean "moderately heritable trait"?

Line 252: The calf doesn't experience dystocia. Dystocia is a condition of the cow.

Line 253: "Bila et"

Lines 253-255: You say here that your results compare with the results from Bila et al. However, in lines 65-66, you said no one has looked at the relationship of body conformation with stillbirth, dystocia, gestation length, or calf birth weight. Aren't these two statements in direct opposition to each other? In one, you say no one has done it and then you say that Bila et al. has compared body conformation to incidence of dystocia.

Line 260: Strong conclusion to draw from such a weak correlation. It may be a significant correlation but it is still extremely weak.

Lines 266-271: Once again, how is mastitis related? You have an in depth explanation of mastitis but no, or at best a very weak, link to your study.

Lines 275-277: Needs rewording, especially the end.

Lines 280-281: Once again drawing a strong conclusion from a very weak correlation.

Lines 283-284: Aren't the ranges the CS, based on lines 111-113? Obviously, CS differs between cows in different CS ranges. Was CS supposed to be calf birth weight?

Line 289: body condition of calves? When was this measured? Where was this discussed?

Lines 302-304: Unclear what you are trying to say here.

Lines 306-307: This is a very strong statement to state here based on the weak evidence presented in this draft of the manuscript. You have weak correlations and did not show any significant regression coefficients (you didn't show regression coefficients period although that was part of your data analysis). The best you can say from the data you have presented is that there is an effect of SRL and RAH on calf birth weight. No body conformation traits affected gestation length based on Table S4, although BD, SRL, and RAH were getting close.

Lines 312-314: When did you score the calves? This is written as if the SRL and RAH were measured on the calves.

Lines 314-317: Very strong statement based on the data presented. You have no data looking at predicting birth traits. You have data showing very weak correlations with gestation length and calf birth weight. You have data showing no differences between gestation length for the different body conformation trait scores.

Would be interesting to see a table like 4 and S4 for dystocia and stillbirths. What are the risks of stillbirth and dystocia for each of the linear score values for each body conformation trait?

There is an inconsistent usage of the Oxford comma (the last comma before "and" in a list). Either use it all the time or none of the time. My personal preference is to use it all of the time.

Reviewer 2 Report

This study aimed to investigate the relationship between dam body conformations and various birth traits in Chinese Holstein calves, with the objective of providing insights for enhancing cow reproductive performance. The authors collected data on multiple conformation traits and birth traits and conducted an analysis to assess the impact of dam conformation on stillbirth, dystocia, gestation length, and calf birth weight.

The findings of the study indicate that specific conformation traits, such as feet angle, set of rear legs, fore udder attachment, and rear attachment height, significantly influenced the risk of stillbirth. Furthermore, an increase in stature and pin width was associated with a decrease in the risk of dystocia. Positive correlations were observed between bone quality, fore udder length, rear attachment height, and gestation length. Additionally, a significant positive correlation was found between fore udder length and calf weight at birth. Calves with higher body conformation traits comprehensive scores exhibited significantly higher birth weights compared to those with lower scores.

These results highlight the importance of improving body conformation traits, particularly the selection of mammary system and overall body shape, in order to enhance the reproductive performance of dairy cows. By considering these factors, breeders and farmers can potentially reduce the risk of stillbirth, dystocia, and other complications during calving, while also promoting optimal calf birth weights.

Overall, this study provides valuable insights into the association between dam body conformations and various birth traits in Chinese Holstein calves, contributing to the knowledge base for optimizing reproductive performance in dairy cows.

Use the term dairy farm instead of cattle farm

Please rewrite lines 47-48, they are not clear

lines 54-55: I would like to suggest that you also discuss the survival of the cow during calving in your study. The survival of the dam is an important aspect to consider when evaluating the impact of body conformations on reproductive performance. Assessing the relationship between dam body conformations and the likelihood of survival during calving can provide valuable insights into the overall health and welfare of the cow.

By including this discussion, you can shed light on whether specific conformation traits influence the cow's ability to successfully give birth and survive the calving process. This information would be beneficial for dairy farmers and breeders, as it can help guide breeding and management practices aimed at improving both calf and dam welfare.

Lines 70-71: Body condition score assessment evaluates the energy reserve of dairy cows and is related to their energy balance status and reproductive ability. Therefore, discussing the survival of the cow at calving will provide a more comprehensive understanding of the impact of body conformations on reproductive performance. You can explore how cows with different body conformations may experience different survival rates at calving and how this may influence the overall reproductive success of the herd. Please see and cite: 10.3389/fvets.2023.1141286

Body conformation traits play a significant role in evaluating the overall muscularity and body condition of dairy cows. By incorporating linear scoring methods, you can assess specific traits and their influence on reproductive performance. Additionally, the use of Legendre Polynomials as a modeling approach allows for a more comprehensive evaluation of muscularity and body condition, taking into account the nonlinear relationships between these traits and reproductive outcomes. See and cite: 10.1080/1828051X.2022.2032850 at line 75.

I believe that further expansion of the Materials and Methods section would greatly enhance the clarity and comprehensiveness of your research. I would like to suggest including the following details:

  1. Measurement of birth traits: Please provide a clear description of how the birth traits (such as stillbirth, dystocia, gestation length, and birth weight) were measured or recorded. Specify the methods used and any standardized protocols followed to ensure consistency across the data collection process.

  2. Number of farms and sample size: It would be beneficial to mention the number of farms from which data were collected for this study. Additionally, provide a justification for the selection of 1,400 dairy cows as the sample size. Explain any criteria used for inclusion or exclusion of cows in the study to help readers understand the representativeness of the sample.

  3. Farm management conditions: Describe the management conditions of the farms included in the study. This may include information on housing systems, feeding practices, health protocols, and other relevant factors that could potentially influence the birth traits and reproductive performance of the cows.

  4. Parity of cows: Specify whether the cows included in the study were primiparous (first calving) or pluriparous (multiple calvings). If both categories were included, discuss how the parity status was taken into account during data analysis to address any potential confounding effects.

  5. Measurements in calves: Provide details on any measurements or assessments conducted on the calves, such as body weight, health parameters, or survival rates. If survival rates were recorded, discuss how this information was incorporated into the analysis and its relevance to the study objectives.

  6. Inclusion of days in milk: If the days in milk were considered as a variable in the analysis, mention how this information was collected and utilized. Discuss the rationale for including this parameter and its potential impact on the observed birth traits.

By including these additional details in the Materials and Methods section, you will provide readers with a clearer understanding of the study design, data collection procedures, and relevant contextual factors. This will enhance the transparency and robustness of your research.

In addition to the previous suggestions, I would like to propose the inclusion of a descriptive statistics section to provide a comprehensive overview of the population included in your study. This will help readers better understand the characteristics and distribution of the sample.

Specifically, I recommend incorporating the following information:

  1. Basic demographic data: Provide the mean, median, and range of relevant demographic variables such as age, parity, and days in milk for the cows included in the study. This will give readers an understanding of the age and reproductive history of the cows in the sample.

  2. Body conformation traits: Present the distribution of the body conformation traits that were assessed using linear scoring or Legendre Polynomials. This can be done by reporting the mean scores, standard deviations, and range for each trait. Including this information will allow readers to grasp the variability and distribution of the body conformation traits in the population.

  3. Birth traits: Describe the distribution of the birth traits (stillbirth, dystocia, gestation length, and birth weight) by reporting relevant statistics such as means, standard deviations, and ranges. This will provide a clear picture of the range and variability of these traits within the population.

  4. Farm characteristics: If possible, provide a summary of the key characteristics of the farms included in the study, such as herd size, management practices, and geographic location. This information will offer insights into the diversity of farm conditions and their potential influence on the study outcomes.

By including a descriptive statistics section, you will provide readers with a better understanding of the population under study and the variability in the measured variables. This will enhance the interpretation and generalizability of your findings.

I have reviewed Table 4 in your manuscript, and I would like to suggest providing additional information to improve the clarity and understanding of the table. Currently, some details are unclear, such as the type of correlation, indexes, and units used in the analysis.

To enhance the reader's comprehension, I recommend including the following information in Table 4:

  1. Type of correlation: Specify whether the correlations reported in the table are Pearson's correlations, Spearman's correlations, or any other type of correlation coefficient. This will help readers understand the nature of the associations between variables.

  2. Index or scoring system: If any specific index or scoring system was used in the analysis, provide a brief explanation or reference to understand its calculation. This will help readers interpret the results appropriately.

  3. Units of measurement: Clearly indicate the units of measurement for each variable included in the table. For example, if birth weight is reported in kilograms, gestation length in days, or body conformation traits on a particular scale, it is important to specify these units for accurate interpretation.

By including these additional details in Table 4, readers will have a better understanding of the analysis and be able to interpret the results more effectively.

Use the term udder instead of breast and nipple

I would like to suggest including some considerations in your manuscript regarding the use of beef bulls on dairy cows. This topic is relevant and can provide valuable insights to readers. Here are a few points to consider:

  1. Reproductive performance: Discuss the potential impact of using beef bulls on the reproductive performance of dairy cows. This may include aspects such as conception rates, calving difficulty, and postpartum reproductive health. Consider referencing relevant studies that have investigated these parameters.

  2. Calf characteristics: Explore the effects of using beef bulls on the characteristics of the resulting crossbred calves. This could include traits such as growth rate, carcass quality, and meat characteristics. Highlight any advantages or disadvantages compared to purebred dairy calves.

  3. Genetic selection: Discuss the importance of appropriate genetic selection when using beef bulls in dairy herds. Emphasize the need to choose beef sires that complement the desired traits of the dairy cows, taking into account both milk production and beef quality.

  4. Economic considerations: Include a discussion on the economic implications of using beef bulls on dairy cows. Consider factors such as the market value of crossbred calves, potential changes in farm revenue, and the cost-effectiveness of this breeding strategy.

  5. Animal welfare: Address the potential welfare implications for both the dam and the resulting calves when using beef bulls in dairy herds. Discuss any considerations for minimizing calving difficulties, ensuring good maternal care, and providing appropriate management practices for crossbred calves. See and cite: https://doi.org/10.1016/j.rvsc.2023.03.008

By incorporating these considerations, your manuscript will provide a more comprehensive analysis of the use of beef bulls on dairy cows and its implications on various aspects such as reproductive performance, calf characteristics, genetic selection, economics, and animal welfare.

In terms of practical implications, it would be beneficial to highlight how the findings of your study can be applied in dairy farming practices. Specifically, discuss how improving body conformation traits, particularly the selection of mammary system and body shape total score, can contribute to enhancing the reproductive performance of dairy cows. Emphasize the potential benefits for dairy farmers, such as improved calf birth weight and reduced risks of stillbirth and dystocia. Providing practical recommendations based on your study's findings can assist farmers in making informed decisions regarding breeding and management strategies.

Additionally, it is important to acknowledge the limitations of your study. Address any potential confounding factors, sample size limitations, or other constraints that may have influenced the results. This will help readers understand the scope and generalizability of your findings and encourage further research in this area.

Including a discussion on the practical implications and limitations of your study will enhance the overall value and applicability of your research findings. Thank you for considering these suggestions.

Author Response

请参阅附件

Round 2

Reviewer 1 Report

In the title, the simple summary, and in a couple of other spots, you refer to the "birth traits of calves" but dystocia and gestation length are more traits of the cow than of the calf.

Line 45: "as well as dam survival is impaired".

Lines 47 and 50: When you modified the text, you messed up the citations. In line 47, you have "[6]. [7,8].[9]" and, in line 50, you have "[7]. [10]"

Lines 62-63: "10 d"

Lines 77-78: "conformation traits were" and Angularity and Heel depth are still unnecessarily capitalized.

Line 79: Consider "moderately" instead of "medium". Moderately is much more commonly used than medium.

Lines 82-83: "has not been well reported to date, with only one study evaluating the impact of body conformation on dystocia [29]."

Line 91: "[22]" should be before the period, not after.

Line 95: Delete "genetic".

Lines 116: "2020 to 2021"

Lines 117-131: 4 spots: "30 to 180 d"

Line 128: Cows don't have litters.

Line 134: Duplicated citation of #25.

Line 154: "260 d"

Lines 155-157: How long is too long? Use earlier edit here too. Also, here you mention that gestation length is calculated from the artificial insemination date but in the introduction, you have that technicians estimate the expected delivery date from a rectal exam.

Lines 157-158: "280 d", "3 mo", "6 d"

Lines 176-177: Abbreviations still don't make sense based on description. Why does VG stand for excellent instead of very good? G, VG, and EXC would make more sense. Did you have any cows that fell below G?

Line 184: Should have a space before "(OR)"

Lines 185-186: Consider deleting "the calf was" and "the cow had"

Line 186: Use "dystocia" instead of "difficult delivery"

Lines 186-188: This is an incomplete sentence. What about the ratio?

Lines 195-201: This should be before you discuss the classification levels of the comprehensive score. Also from lines 173-175, what's the difference between composite score (line 173, 177) and comprehensive score (line 175, 195, 197)? For line 197, if you've already defined composite/comprehensive score, CS should be used here instead of being written out.

Table 4 and description: Still not sure how this shows that there is an impact on stillbirth by FA, SRL, FUA, and RAH. Most of the P-values are greater than 0.05 and it seems like there is no real pattern as to which one's are different. There has got to be a better way to show this to determine if these traits actually affect the probability of having a stillborn calf. For the footnote for Table 4, you state that the reference level is for the lowest score of that trait in the herd. For FA and FUA, you are comparing to 4 but there are cows that scored 2 and 3 points.

Line 252: Where does the P < 0.05 come from? Most the P-values presented in Table 4 are greater than 0.05.

Line 255: Semicolons are not followed by capital letters. Would be more appropriate to use a period instead of a semicolon here.

Line 258: Should be a period instead of a semicolon.

Line 259: Missing a period at the end of the sentence.

Table 5 and description: I'm not sure you know what you are describing here.

Line 278: Where does the P < 0.05 come from? You have P-values greater than 0.05 in Table 5.

Table 6: You changed in your Materials and Methods to using Pearson correlation instead of Kendall rank correlation but, here in this table, it still says Kendall rank. Since the correlation values changed, I assume you actually did change to Pearson.

Line 300: Repeat of what was written in line 291. As a result, I think section 3.3 and 3.4 need to be combined or you need to make sure that you are only discussing gestation length in one section and birth weight in the next.

Lines 340-370: Still not sure how your results support what you state here. P-values over 0.05 do not show a linear relationship of any of the traits mentioned with the probability of stillbirths.

Lines 378-382: Poor English structure to this sentence.

Line 395: Incomplete sentence that has two periods at the end, one before the reference and one after the reference.

Line 396: References without a sentence.

Line 426: Consider "small" instead of "little".

Line 453: "crossbreeding between beef cattle and cows" - What type of cows?

Lines 468-489: Still drawing some pretty strong conclusions from weak data.

References: Inconsistent formatting throughout.

There is still an inconsistent usage of the Oxford comma (the last comma before "and" in a list). Either use it all the time or none of the time. My personal preference is to use it all of the time.

Reviewer 2 Report

after the revisions the paper improved a lot, I endorse the publication

Author Response

Dear Reviewer,

Thank you for taking time out of your busy schedule to read and revise our manuscript, and your valuable comments on our manuscript have made our manuscript more perfect. Thank you very much for your approval of the revised manuscript.

Best regard

Jiayu Yang